# Systematic Review of Electricity Demand Forecast Using ANN-Based Machine Learning Algorithms

**DOI:** 10.3390/s21134544

**Published:** 2021-07-02

**Authors:** Antón Román-Portabales, Martín López-Nores, José Juan Pazos-Arias

**Affiliations:** 1Quobis, 36380 O Porriño, Spain; 2atlanTTic, Universidade de Vigo, 36310 Vigo, Spain; mlnores@det.uvigo.es (M.L.-N.); jose@det.uvigo.es (J.J.P.-A.)

**Keywords:** electricity demand forecast, machine learning, artificial neural networks, systematic review

## Abstract

The forecast of electricity demand has been a recurrent research topic for decades, due to its economical and strategic relevance. Several Machine Learning (ML) techniques have evolved in parallel with the complexity of the electric grid. This paper reviews a wide selection of approaches that have used Artificial Neural Networks (ANN) to forecast electricity demand, aiming to help newcomers and experienced researchers to appraise the common practices and to detect areas where there is room for improvement in the face of the current widespread deployment of smart meters and sensors, which yields an unprecedented amount of data to work with. The review looks at the specific problems tackled by each one of the selected papers, the results attained by their algorithms, and the strategies followed to validate and compare the results. This way, it is possible to highlight some peculiarities and algorithm configurations that seem to consistently outperform others in specific settings.

## 1. Introduction

Electricity is expected to increase its prevalence as the main energy vector in the near future for industrial, domestic and transportation use. This emphasizes the importance of electricity demand forecast, as it has a direct impact on many operational and business processes. Electricity demand is typically known as *load* in the electrical engineering jargon, we will use both terms interchangeably. For decades, load forecast has been a recurrent research topic and a framework for the evolution of Machine Learning (ML) approaches based on Artificial Neural Networks (ANN), which are inherently suitable to deal with non-linearities and multiple types of inputs [1,2]. Presently, the massive deployment of smart meters and sensors along the grid yields a propitious environment for the optimization of such techniques.

The literature accumulated on the topic of load forecast using ANN-based models over the last 20 years is vast and difficult to grasp. This paper aims at classifying and reviewing the most relevant works. Our focus is on identifying what algorithm performs better for specific electricity demand problems and under what circumstances, including the selection of input variables and the optimal combination of parameters. Other distinguishing aspects of this systematic review are the following:We analyze the Key Performance Indicators (KPIs) used to evaluate the accuracy of the predictions and to compare the performance of different algorithms. In this regard, the predominance of some metrics in the literature (e.g., MAPE, the Mean Absolute Percentage Error) often leads to overlooking important quality parameters, such as the distribution of the error and the maximum forecast error.We look at other fundamental aspects in ML problems, such as the data pre-processing techniques, the selection of training and validation sets, the tuning of the hyper-parameters of the model, the graphical representations and the presentation of the results.Last but not least, we discuss the ability to publicly access the datasets used to carry out the experiments and to validate the results and the code of each one of the selected papers. Lack of access makes the results of many papers very hard or impossible to reproduce, reducing their impact as sources of innovation and knowledge.

Previous reviews of approaches for electricity demand forecast (see [3,4,5]) surveyed the use of ANN-based techniques in a shallower manner, as they covered other ML techniques too. Other surveys looked at general uses of ML in energy systems, not only for load but also for generation, and not restricted to electricity but considering any sources of energy [6,7]. Our exclusive focus on ANN for electricity demand forecast allows providing deeper insight, to the point of questioning aspects that have been traditionally taken for granted, such as the non-linear nature of the forecast problem (to be discussed in Section 4.3.3). It is worth noting, though, that we cover not only pure uses of ANNs, but also hybrid approaches in which ANNs are combined with other algorithms and/or used to process the data in early or final stages.

## 2. Methodology

Initially, we used Elsevier’s ScienceDirect, Scopus and IEEE Xplore to search for relevant papers, thus ensuring essential quality requirements and coverage of the most relevant publications. We obtained an initial list by performing search queries for the keywords “ANN”, “neural networks”, “forecast”, “prediction”, “electricity”, “load”, “forecasting” and “machine learning”. We also considered related papers that were recommended by the search engines and met the search requirements. Next, we left out all the papers that did not include ANN-based mechanisms or dealt with other energy sources than electricity —still, we included papers that compared ANN-based methods to other approaches such as Support Vector Machines (SVM). We proceeded iteratively to include all the papers referenced in the state-of-the-art section of papers already included in our set.

Table 1 shows the sites from where we downloaded the papers covered in the review. 55% of them were retrieved from IEEE Explorer, acknowledging the fact that many relevant papers on electric load forecasting papers have been traditionally presented in IEEE conferences. MDPI and ScienceDirect also hosted a relevant number of original papers.

Having selected the papers, we put them on a data sheet with different columns to look at the specifics of each one. The columns were:Type of problem to solve.Algorithms used.Supporting tools.Input variables.Dataset characteristics.Performance indicators.Results.Particularities.

In the last column we wrote down comments about what made each paper different from others. This helped us to analyze and compare the different papers focusing on specific aspects that we will cover in the review. A simplified version of this table is included in Section 5 to be used as a quick reference by the readers.

## 3. State-of-the-Art ANN-Based Algorithms Used in Load Forecasting Problems

Some of the reviewed papers use single ANN-based algorithms, whereas others combine them with other techniques. The single algorithms are the following:The Multi-Layer Perceptron (MLP) refers to a canonical feedforward artificial neural network, which typically consists of one input layer, one output layer and a set of hidden layers in between. Early works showed that a single hidden layer is sufficient to yield a universal approximator of any function, and so MLPs were commonly used in papers from the 1990s and early 2000s. However they have been progressively replaced by more sophisticated recursive algorithms, which can better capture the complex patterns of load time series. The most recent papers included in Section 5 show how recursive ANN-based approaches typically outperform MLP.Self-Organizing Maps (SOM) are neural network-based dimensionality reduction algorithms, generally used to represent a high-dimensional dataset as a two-dimensional discretized pattern. They are also called *feature maps*, as they are essentially re-training the features of the input data and grouping them according to similarity parameters. SOMs are used to recognize common patterns in the input space and train distinct ANNs to be used with the different patterns [35].Deep Learning refers to ANN networks capable of unsupervised learning from data that are unstructured or unlabeled. The adjective “deep” comes from the use of multiple hidden layers in the network to progressively extract higher-level features from the raw input.Many authors (e.g., [20,47,52]) use variants of *Recursive Neural Networks* (RNNs) that have the capability of learning from previous load time series. Others use *Long Short-Term Memory (LSTM*) networks, a special kind of RNNs that can learn from long-term dependencies. These were introduced by Hochreiter and Schmidhuber [57] in 1997 and refined and popularized by many authors in subsequent works. Several of the most recent papers included in the review conclude that LSTM variants achieve low forecasting errors outperforming other algorithms in their experiments.

The hybrid ANN-based algorithms found in the reviewed papers fall into three approaches:ANN and Genetic Algorithms (ANN-GA). In these works, the idea of the genetic algorithms is to iteratively apply three operations (referred to as selection, crossing and mutation) in order to optimize different parameters of the ANNs. For example, Wang et al. [33] used the GA to improve specifically the back-propagation weights, whereas Azadeh et al. [36] used GAs to tune all the parameters of an MLP.ANN and Particle Swarm Optimization (ANN-PSO). PSO is another optimization technique that tries to improve a candidate solution in a search-space with regard to a given measure of quality. It is a metaheuristic (i.e., it makes few or no assumptions about the problem being optimized) that can search very large spaces of candidate solutions, but it cannot guarantee that an optimal solution is ever found. As an example, Son and Kim [58] used PSO to select the 10 most relevant variables to be used as input for SVR (Support Vector Machine Regression) and ANN algorithms. Likewise, He and Xu [22] proposed the use of PSO to optimize the back-propagation process to tune the parameters of an MLP.Adaptive Neuro-Fuzzy Inference System (ANFIS). Developed in 1993 by Jang [59], ANFIS overcomes the deficient parts of ANNs and fuzzy logic by combining both technologies. It is used in [3] to model load demand problems. It uses fuzzy inference in its internal layers which allows the model to be less dependent on proficient knowledge, improving its learning and making it more adaptable.

Recent papers combine at least two ANN-based algorithms. For instance, Ref. [32] integrates LSTM with Deep Neural Networks (DNN) to forecast load demand from previous time series and to predict from meteorological input variables. In this case, LSTM captures the load forecast due to previous values thanks to its recursion features, and the DNN gives a more accurate value for the load demand specifically owing to the weather conditions.

## 4. Particularities of Electric Load Demand As a Problem for ANNs

In this section, we shall highlight particular aspects about the use of ANNs for load forecasting. These are questions that must be taken into account in any research work, as they condition the type of algorithms that may be used.

### 4.1. Prediction Range

According to the time range of the prediction, we can distinguish three categories that have been used in the definition of energy forecast problems, at least since 1995 [10]:Short-term load forecasting (STLF) refers to predictions up to 1 day ahead.Medium-term load forecasting (MTLF) refers to 1 day to 1 year ahead.Long-term load forecasting (LTLF) refers to 1–10 years ahead.

Table 2 shows that most of the reviewed papers that use ANN-based algorithms do so for STLF problems. Therefore, we can safely assume that ANN-based algorithms have been widely recognized as suitable for short-term prediction.

STLF has become particularly important (hence the greater presence in the scientific literature) since the massive introduction of renewable energy sources, as the forecasts help the electric companies to plan the production mix more efficiently. STLF is crucial for electric intra-day markets, where 1-day ahead forecasts are used to fix the prices for the next day considering the expected demand. STLF is also important for the operation of electric companies and microgrids, where the predicted demand may drive operative decisions in order to be properly covered by the generation sources. Many electric operators are supporting these research efforts by providing significant amounts of data and funding.

ANN-based algorithms have been also proven to work well for MTLF when they can capture weekly and seasonal patterns, as it happens with the recursion techniques of LSTM [43]. LTLF problems, in turn, seem harder to solve by using ML algorithms only. The expected demand in the next years depends heavily on demographic, geopolitical and technological evolution variables, which are hard to turn into numbers and for which there are no historical data to learn from.

### 4.2. Load Forecasting as a Sequence Prediction Problem

In the electricity forecast field, sequences are typically series of past ordered load values, indexed by time. Brownlee [60] differentiated two types of prediction problems:Sequence prediction: from a sequence of values a single value is predicted. For example, from a time series of previous load values we obtain a prediction for the next load value.Sequence-to-Sequence (S2S) prediction: we do not obtain a single value but a sequence of predicted values, defining how the load will evolve in a range of future time steps.

Our review covers papers featuring both approaches, and even combined strategies (e.g., [19]).

### 4.3. Input Variables

In many cases the selection of input variables is determined by the available data. All the papers covered in this review consider the previous load (directly or applying some kind of transformation) as one of the input variables of the ML algorithm. In many cases, a time series of previous load is the only input to the algorithm, which is required to learn from past values only. In other cases, it is common to use additional data such as weather variables and economic activity indicators [29,33,58]. Table 3 shows the distribution of the input variables used in the analyzed papers.

Weather variables—especially temperature—are known to have a linear influence on the electricity demand [47]. Extensive analyses of the influence of weather variables, daylight hours and human activity can be found in [18,61]. It has been shown (see [33]) that the load data over the same period or previous periods have greater influence, though, as those values of electric load implicitly capture effects of climate, daylight hours and human habits.

In the electricity market, real-time price depends on the generation of renewable sources, current demand and socio-economic factors. Real-time price has not been considered as an input variable for STLF problems in the reviewed papers. However it exhibits a relevant non-linear correlation with power load as shown in [62].

The values provided by the Advanced Metering Infrastructure (AMIs) deployed by electric companies give the amount of energy consumed during a period of time (typically 1 h and 24 h) but there are also sensors that can provide instantaneous values of consumed power. They are all valid for the predictions, but energy values in KW/h or W/h are the most commonly used in forecasting problems. The AMIs can also provide the peak values directly and in many cases the forecasting is focused on the peak values only, not on aggregated consumption.

To identify the predictive value of input variables, several authors have made correlation analysis between the different input variables and the power load [18,42]. However, recent papers propose more advanced alternatives such as Kendall rank and Copula functions [62,63] which are more suitable to identify non-linear relationships between input and output variables. The use of LASSO (Least Absolute Shrinkage and Selection Operator) regression analysis to select the most relevant features is also proposed as an effective approach [45]. This technique enables the selection of the features to optimize the performance of the model. Feature selection is important to improve the performance of models with a huge number of features, typically those which include socio-economic variables.

#### 4.3.1. Sources of Input Data

All the reviewed papers used time series of previous electric demand to train and test the models. Table 4 shows the sources of the data.

Many of the papers focused on certain geographic areas, so they handled problems of aggregated demand from thousands or millions of consumers. The use of ANN-based models to these problems has shown very good performance. The demand prediction problems using smart meter and microgrid data, in turn, seem to be in an early stage of evolution, as they handle load patterns whose distributions differ significantly from those of aggregated demands.

Several studies have proved that forecast is much more accurate when it is done over aggregated data. For example, Kong et al. [31] propose the use of a clustering technique called DBSCAN (Density-Based Spatial Clustering of Application with Noise) to evaluate consistency in daily power profile, finding that aggregated data presents fewer outliers, which favors ANN convergence. The same authors compared the forecast accuracy of individual meters and checked how it improves with the level of aggregation, discovering that the aggregation of forecasts is more accurate than the forecast of the aggregation. Regarding the patterns of individual consumers, lifestyles are clearly reflected in energy consumption as consumers typically have common and repetitive behaviors [64].

#### 4.3.2. Pre-Processing of Input Variables

The importance of data pre-processing is a well-known topic in data science [65]. Any forecasting problem requires processing of data before feeding them to whichever ML algorithm. However, most of the papers covered in this review do not explain the way they pre-process the numeric data.

The pre-processing may differ depending on the used algorithm, but it will typically involve the following steps when using ANN-algorithms:Data cleaning. Either due to errors in the sensors or in the data processing, the time series may include invalid or missing data, making it necessary to apply conventional mechanisms to modify these values. For example, depending on the type and amount of missing data, different approaches can be used, such as dropping the variable or completing with the mean or the last observed value. Removal of duplicate rows may be also needed at this initial stage. Very few papers explain whether any of these techniques was used, even when they may have a significant effect on the model’s performance.Data validation. It is necessary to validate the data, especially when they come directly from AMI devices and they have not been obtained from public databases. Data visualization techniques can help to check if the data match expected patterns. It is worth noting that smart meters typically send the measured values using PLC (Power Line Communication) technologies, which may be affected by different electromagnetic interference sources [66]. This makes it especially important to verify the integrity of the data before training the model. Detection and removal of outlier values is typically performed to optimize the training of ANN-based models. Ref. [46] proposes the use of PCA (Principal Component Analysis) as an effective outlier detection approach.Data transformation. This phase includes different types of transformations of the data, such as change of units or data aggregation. Data aggregation from individual sources is a common practice to achieve data reduction, change of scale and minimize variability. More advanced transformation techniques are aimed at rescaling the features in order to make the algorithm to converge faster and properly and minimize the forecasting error. The most common rescaling approaches are normalization and standardization:
Normalization refers to the process of scaling the original data range to values between 0 and 1. It is useful when the data have varying scales and the used algorithm does not make assumptions about their distribution (as is the case of ANNs).Standardization consists of re-scaling the data so that the mean of the values is 0 and the standard deviation is 1. Variables that are measured at different scales would not contribute equally to the analysis and might end up creating biased results through the ANNs. Standardization also avoids problems that would stem from measurements expressed with different units.Dimensionality reduction techniques are typically used in machine learning problems in order to optimize the model generation by reducing the number of input variables. However, only a few of the reviewed papers require the usage these techniques due to the low number of input variables.

The blue box in Figure 1 presents a generalization of the pre-processing steps found in the reviewed papers. As already mentioned, many authors do not explain how the raw data are pre-processed, even though any omission or error in this process may lead to inaccurate and suboptimal models.

#### 4.3.3. Non-Linearity with Respect to Input Variables

In almost all the reviewed papers, the authors mention the fact that electricity demand is inherently non-linear, and therefore algorithms designed for linear problems are not a good choice for forecasting. This is typically taken for granted, without referring to papers that include mathematical analyses of demand time series in order to calculate the degree of linearity regarding the input variables. In this line, Darbellay and Slama [34] carried out a correlation analysis that suggests that LTLF—at least with the data available from the Czech Republic—was primarily a linear problem. This was confirmed by the comparison of the predictions. Knowing that, the same authors discussed the conditions under which ANNs could be superior to linear models. It is relevant to note that the computational cost of ANN-based algorithms can be easily afforded by research centers and companies of any size at present. Therefore, the superior mathematical knowledge required to create adapted linear models may not be worth even when the algorithms are typically lighter than the training and optimization process of ANN-based algorithms.

### 4.4. Output Variables

In the reviewed papers we found two main possible output variables:A time series of expected demand for the future, i.e., a list of the demand values predicted for specific moments.The load peak value of the electric grid at some point in the future (e.g., next day or next week peak).

As shown in Figure 2, the most common output is the 24-h ahead prediction. As we explained before, this is especially relevant because the production is scheduled according to the negotiation of the intra-day electricity markets.

Figure 2 also shows that the number of papers that look at peak values only ([8,37,56]) is very low compared to those that predict the load time series, and none of those was published after 2011. Narrowing to peak values was apparently done to simplify the problem, but currently predicting a complete time series is more useful for operative purposes (and, of course, peak values can be drawn from the predicted time series).

Almost all the reviewed papers present models that generate a predicted value for future points in time. Other authors [67] propose models that produce forecast probability density functions (PDF) so that it is possible to know the degree of uncertainty of each result. The suitability of a PDF as the output of the model depends on the type of problem that needs to be solved. A PDF may provide valuable information, for example, when it is important to know the probability of rare events that can compromise the supply. The loss of information with single-point forecasting is especially relevant when the forecasting model produces fat-tailed PDFs [68].

### 4.5. Measuring and Comparing Performance

The reviewed papers typically used the same data set with different algorithms or variants to decide which one performs better. Several Key Performance Indicators (KPI) have been used in order to compare their results.

Most of the works compare the results of the simulation algorithm with the actual values. The most common metrics to do so is the Mean Absolute Percentage Error (MAPE), given by Equation (Equation 1), where *N* represents the number of predicted values, $F_{t}$ is the predicted value at *t* and $A_{t}$ is the actual value which corresponds to the predicted value. MAPE gives a measurement of how accurate the prediction is, based on the average percentage of error of each predicted value.
(1)$$\mathrm{MAPE}=\frac{1}{N}\sum_{t=1}^{N}\left|\frac{A_{t}-F_{t}}{A_{t}}\right|$$

The Mean Absolute Error (MAE), given by Equation (Equation 2), is equivalent to the MAPE but gives an absolute value for the error rather than a percentage.
(2)$$\mathrm{MAE}=\sum_{t=1}^{N}\left|\frac{A_{t}-F_{t}}{A_{t}}\right|$$

When the same dataset is used to compare the prediction algorithms, both MAPE and MAE can be used; however, they are not helpful to compare results from different datasets. Even with the same dataset, the use of MAE may lead to confusing results if the units of any output are modified. Thus, MAPE is more common in the reviewed papers.

The second most common KPI is the Root Mean Square Percentage Error (RMSPE), given by Equation (Equation 3). While the MAPE gives the same weight to all errors, the RMSPE penalizes variance, since it gives more weight to larger absolute values than errors with smaller absolute values. Like in the case of the MAE, there is an absolute version called RMSE (Equation (Equation 4))that gives more weight to larger errors.
(3)$$\mathrm{RMSPE}=\sqrt{\frac{1}{N}\sum_{t=1}^{N}{\left(\frac{A_{t}-F_{t}}{A_{t}}\right)}^{2}}$$
(4)$$\mathrm{RMSE}=\sqrt{\frac{1}{N}\sum_{t=1}^{N}{\left(A_{t}-F_{t}\right)}^{2}}$$

RMSPE is considered more suitable to show bigger deviations and helps to provide a complete picture of the error distribution (see [69]); however, it is not commonly used in the analyzed papers. Chai and Draxler [69] claim that RMSE is more appropriate than MAE when the error distribution is expected to be Gaussian, but this is often disregarded in the reviewed papers even though it would help to extract more information from the results.

The following are other variables found in the literature, depending of the purpose of the research work:The Maximum Negative Error (MNE) and Maximum Positive Error (MPE) give the maximum negative and positive difference, respectively, between a predicted value and a real value. These values can be more relevant than the average error for some applications (e.g., to forecast the fuel stockage in a power plant).The Residual Sum of Squares (RSS) is the sum of the squares of residuals (deviations predicted from actual values of data). It can be calculated from the RMSE. It measures the discrepancy between the data and an estimation model.The Standard Deviation of Residuals describes the difference in standard deviations of observed values versus predicted values as shown by points in a regression analysis.The comparison of the correlation between the time series produced by different algorithms and the real validation set is used by some authors to measure quality [13], too.

Computing the values above allows comparing the results attained by different techniques. However, such a simple analysis may not be very meaningful, especially when the difference between algorithms is small or the dataset is not very long. In this line, Kandananond [48] used Wilcoxson signed-rank and paired t-tests to compare the results offered by ANN, MLR and ARIMA. The *p*-values obtained were above α=0.05, so he concluded that the results were not meaningful and there was no real advantage of ANN over ARIMA or MLR.

### 4.6. Forecasting Model Generation Process

Figure 1 shows the steps typically followed by the reviewed papers to validate their proposed models by obtaining meaningful results from the input data. The process is not significantly different from the flows typically used in ML problems.

Once the data have been pre-processed following the steps presented in Section 4.3.2, they are typically divided in two sets: training and test. The training data are in turn divided into training and validation datasets. First, the training data is used to train the model. The initial version of the model is validated with the validation dataset. After this process, it may be necessary to tune the hyperparameters of the model or, in the specific case of ANN-based models, to introduce some changes to the topology of the neural network. Then the model is tested again with the test dataset and its performance is measured by calculating the error with regard to the real data.

Graphical representations of the final results are included in almost all the reviewed papers. It is also common to support model selection decisions with different data visualization techniques to describe the patterns and characteristics of the original data.

Despite the fact that this structure is common to most of the reviewed papers, the authors typically introduce innovations in either the methodology or the algorithms and techniques used in the different stages of the process. The parts where more innovative approaches can be found are:The use of clustering and other categorization techniques with load input data in order to tune the model depending on the patterns found therein [31,35,49,51];The ML algorithm proposed to build the forecasting model. The proposed models are typically compared with commonly used ML models. Section 5 includes an exhaustive list of both the ML algorithm being tested and the alternative options.As mentioned in Section 4.5, authors use different alternatives to analyze and compare the performance of the different ML algorithms [13,40,48]. However, the comparison of MAPE and RMSE values is the most typical choice.

### 4.7. Reproducibility of Forecasting Experiments

As shown in Table 5, we found that less than 40% of the reviewed papers used publicly accessible data that could be used to reproduce the experiment. In the other cases, the researchers typically had some type of agreement with the operator providing the data, and the original data are not accessible. This makes the experiments hard to reproduce and validate, especially in the case of new algorithms. In any case, aggregated demand and generation data are commonly available in developed countries. In contrast, smart meter data are harder to achieve owing to data protection laws, but it is possible to gain access to anonymized load time series of individual and industrial consumers, which can be freely used for experiments.

Another factor that affects the reproducibility of the experiments are the tools and the code used to conduct them. The growing adoption of ML algorithms to extract value from the massive amount of data available in numerous fields of applications has fostered an active open-source ML community. Some of the most relevant ML and data science related projects (e.g., PyTorch, Tensorflow and its high-level API Keras) are supported by big Internet companies. Research in ML can now take advantage of these valuable tools, reducing the programming efforts and making it easier to focus on the problems and try different alternatives. In Table 6, we see that MATLAB remains the main tool used in the reviewed papers, while several authors used code implemented ad-hoc. The tool used for the implementation is not even mentioned by many authors.

Regarding the code used to conduct the experiments, only one of the reviewed papers offers it to the reader [52]. However, sharing the code seems to be a growing trend in data science and ML papers [70] so it may happen as well for load demand forecasting papers in the near future.

## 5. Summary of the Reviewed Papers

Given the perspective of the previous section, next we provide a table (Table 7) containing the most relevant information from the reviewed papers, including the following:Title and reference.Year of publication.Goal.Algorithms and optimization techniques used.Performance of the best algorithm.

Most of the papers used MAPE (and in some cases other related values) as the metrics to compare the performance of the algorithms. To give the reader a reference of the performance of each algorithm, we only include the MAPE value in the table. When other non-normalized values were used, we did not include them in the table to keep it coherent and avoid misunderstandings. If, in some specific case, the MAPE was not the most relevant value, it is indicated in the *Best algorithm* column.

## 6. Conclusions

The use of ANN-based ML algorithms for electricity demand forecasting is an idea that goes back to the 1990s, but continues to be the subject of intensive research presently. Chronologically, the papers we have reviewed show how ANNs evolved from a sensible and promising concept—due to the cyclic nature of load demand—to a widely used reality in production environments.

This review is aimed at providing a valuable asset for researchers. Bearing this goal in mind, we analyzed 50 research works to extract their common patterns and also the main differences in terms of methodology and algorithms. We want to highlight four aspects to be considered for future forecasting experiments: performance comparison, best performing algorithms, influence of aggregation level, and experiment reproducibility.

### 6.1. Performance Comparison

The generalized use of MAPE to measure the performance of the algorithms allows to extract some relevant conclusions. The first is that ANN-based algorithms (and especially LSTM, which is the most used algorithm in the reviewed papers) have proved to achieve very good results in aggregated load forecast, and that their predictions get typically more accurate as the number of electricity consumers grows. A significant number of recent papers show MAPE values below 3% for the best cases.

Regarding the ability to compare the different algorithms, we understand that just comparing the MAPE values from different papers can give a raw orientation for future research works. However, we are also aware that this is not the best approach, since they are making predictions over different datasets, which in many cases are not accessible to the scientific community. Additionally, using the MAPE as the single KPI may not be always fair, since the RMSE may be a better metric for many applications where high forecast errors must be avoided. It is worth noting that recently published papers typically include (at least) both values, which a positive practice to enable more complete comparisons in the future.

### 6.2. The Best Performing Algorithms

ANN-based approaches that can capture recurrent patterns (such as RNN and, specifically, LSTM) proved to perform well for load demand problems. In consequence, most of the papers covered in this survey featured one ANN-based algorithm as the best alternative compared to other approaches. However, there are some exceptions. For instance, in [58] a combination of PSO with SVR turned out to perform slightly better than PSO with ANN-based algorithms; Ref. [43] found the autoregressive models of ARIMA to outperform RNN and LSTM for STLF problems; SVM worked better than MLP in an STLF problem in [3]; and SVM was found to be more reliable and stable than ANN for mid-term load forecasting in [29].

In general, combinations of MLP or LTSM with other algorithms do not attain a substantial advantage over the original algorithms, but the papers that compare innovative combinations typically show them as the optimal option. There are innovative models, though, whose authors claim to obtain MAPE values below 1% [71]. However, without an extensive validation using different datasets, it remains unclear whether the model really shows a very good performance for generic load demand problems, or the results may be due to an over-fitted model (e.g., one that provides very good results only for the dataset with which it has been trained). An alternative to obtain more accurate models—at the cost of a higher complexity—could be the kind of combinations of different ANN-based algorithms as proposed in [32].

### 6.3. Influence of the Aggregation Level in Model Performance

The accuracy of STLF and MTLF predictions for aggregated demand of a huge number of consumers is good in general, which makes modern ANN-based algorithms a good tool for commercial and research purposes. In turn, load forecasting in microgrids is a challenging problem according to the results provided by the analyzed papers. The MAPE results are typically above 10%. Still, this could be good enough, inasmuch as recent advances in energy storage techniques can easily absorb the forecast errors.

The problem of individual user load forecasting seems to be the hardest to resolve, which is understandable due to the nature of some human behaviors. The high MAPE values attained by the few papers that tackle this problem (such as [31]) suggest that ANN may not be the best approach if very high precision is needed. Again, the importance of individual consumer forecast is lower than aggregated load from the point of view of the industry, due to the recent improvements in power storage technologies that can absorb load oscillation in isolated systems. In any case, we understand that there is still room for improvement for microgrids and individual load demand forecast models.

### 6.4. Benchmarks and Reproducibility

To make an unbiased assessment of the performance of the different algorithms, load demand papers should use a common reference benchmark, which does not exist yet. This could use publicly available datasets, in addition to other specific ones. For example, the comprehensive list of smart meter time series included in [73] could be used as a starting point to define a reference dataset to test the performance of the different algorithms in equivalent conditions. In the same line, the publication of results without making the source code and datasets available makes it hard or impossible to reproduce the results. Fortunately, sharing the source code is also becoming common in recent years [70], so we are optimistic in this sense. Without a doubt, this will help to take forecasting closer to the limits of ML techniques in the next few years.

## Figures and Tables

**Figure 1 sensors-21-04544-f001:**
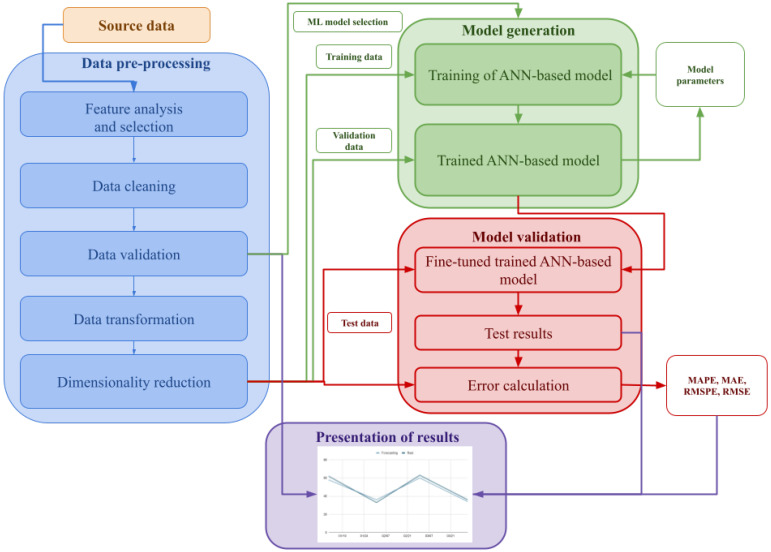
Generalization of the steps documented in the reviewed papers.

**Figure 2 sensors-21-04544-f002:**
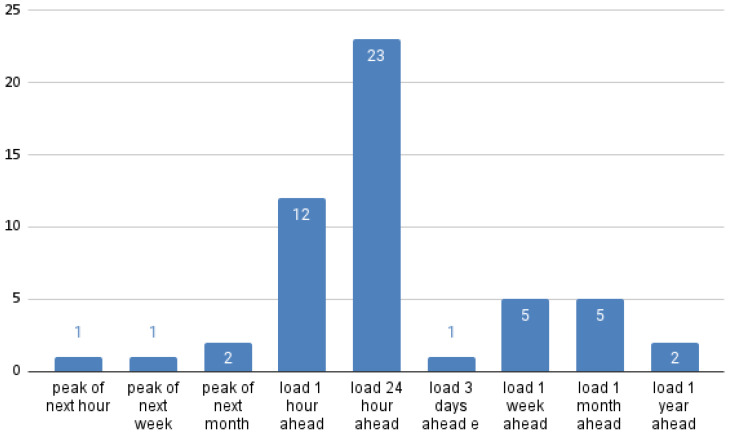
Output variables of the reviewed papers that focus on STLF and MTLF. Some papers are counted in several columns.

**Table 1 sensors-21-04544-t001:** Sources of papers for the review.

Publisher	Number of Papers	References
IEEE	29	[2,3,8,9,10,11,12,13,14,15,16,17,18,19,20,21,22,23,24,25,26,27,28,29,30,31,32,33]
ScienceDirect	12	[34,35,36,37,38,39,40,41,42,43,44,45]
MDPI	8	[44,46,47,48,49,50,51]
Arxiv	3	[52,53,54]
Others	2	[55,56]

**Table 2 sensors-21-04544-t002:** Type of used input variables.

Type of Forecast	Number of Papers
STLF	46
MTLF	8
LTLF	2

**Table 3 sensors-21-04544-t003:** Type of used input variables.

Input Variable	Number of Papers
Previous load time- series	37
Previous load and weather time series	10
Previous load, weather and economic variables time series	3

**Table 4 sensors-21-04544-t004:** Origin of load time-series data.

Data	Number of Papers
Aggregated data from a geographic area	34
Aggregated data from microgrids	8
Individual meters deployed in the public power grid	13

**Table 5 sensors-21-04544-t005:** Data source in the reviewed papers.

Data Source	Number of Papers
Public data	14 ([12,20,24,25,26,31,40,47,50,51,52,53,54,58])
Private data	37

**Table 6 sensors-21-04544-t006:** Tools used in the reviewed papers.

Tool	Number of Papers
Not mentioned	19
MATLAB	12
Tensorflow-based	6
Custom code	3

**Table 7 sensors-21-04544-t007:** Reviewed papers.

Title	Year	Goal	Algorithms	Best Algorithm
An artificial neural network-based short term load forecasting with special tuning for weekends and seasonal changes [8]	1993	To compare the performance of ANN using season, day of week, temperature and previous power peaks as inputs to forecast 1-week ahead peaks.	MLP	MAPE MLP: 1.60%
A recurrent neural network for short-term load forecasting [9]	1993	To compare the performance of recurrent and feedforward ANNs.	Feedforward 3-layer MLP 3-layer recurrent neural network with BP and diffusion learning	MAPE RNN with diffusion learning: 2.07%
Practical experiences with an adaptive neural network short-term load forecasting system [11]	1995	To compare performance of statistical methods and MLP to forecast demand 7 days ahead in blocks of 3 h.	3-layer MLP (hidden layer with 3 neurons) with daily, weekly and monthly adaptation	MAPE MLP: 6%
A real-time short-term peak and average load forecasting system using a self-organising fuzzy neural network [38]	1998	To predict the demand peak 1 day and 1 week ahead comparing the performance of SFNN (Self-organising Fuzzy Neural Network), FFN (Fuzzy Neural Network) and MLP.	SFNN, FFN and MLP	MAPE SFNN: 1.8% for 1 day ahead peak load forecast and 1.6% for 1 week ahead
Forecasting the short-term demand for electricity: Do neural networks stand a better chance? [34]	2000	To compare feedforward ANN with ARIMA and ARMAX using previous demand and temperature as inputs. To analyze the non-linearity of the demand forecast problem.	ARIMA, ARMAX and MLP	MAPE MLP: 0.8%
Global model for short-term load forecasting using artificial neural networks [12]	2002	To check performance of MLPs trained for classes defined using self-organizing maps with statistical methods. No comparison with other algorithms.	Kohonen’s self-organising map + Elman Recurrent Network	MAPE: 1.15–1.61%
A new approach using artificial neural network and time series models for short term load forecasting [13]	2003	To check accuracy of ANN to predict forecast using input variables selected depending on their correlation coefficient compared with ARIMA.	MLP using correlation coefficient to calculate weights	MAPE: 2.241%
Forecasting electrical consumption by integration of Neural Network, time series and ANOVA [39]	2007	To compare the performance of MLP to predict aggregated load from time series using analysis of variance and time series approach. Linear regression ANOVA and Duncan’s Multiple Range Tests are used to validate results.	MLP	MAPE: MLP 1.56%
Integration of artificial neural networks and genetic algorithm to predict electrical energy consumption [36]	2007	To check performance of MLP and GA for LTLF in the Iranian agricultural sector.	MLP + GA	MAPE MLP: 0.13%
Annual electricity consumption forecasting by neural network in high energy consuming industrial sectors [40]	2008	To check the performance of ANN algorithm to predict annual load of energy intensive industries using different input variables such as electricity price, number of consumers, fossil fuel price, previous load and industrial sector. ANOVA and Duncan’s multiple range test are used for formal comparison and validation.	MLP using different networks and regression.	MAPE: MLP 0.99%
Daily load forecasting using recursive Artificial Neural Network vs. classic forecasting approaches [23]	2009	To compare the performance of RNN with other analytical methods for 24-h ahead forecasts for a region of Romania.	RNN (using hyperbolic tangent as activation function).	RNN performs better. Least square value used instead of MAPE.
Short-term load forecasting using artificial neural networks [24]	2009	To compare the performance of ANN for 1-h ahead performance using previous load, weekday, month and temperature as input values with the results of other studios. ISO-New England control data are used to validate the algorithm.	Feed-Forward MLP using LM as BP algorithm.	MAPE: 0.439% (for ISO-New England)
Dynamic neural network-based genetic algorithm optimizing for short term load forecasting [33]	2010	To compare BP and Genetic Algorithm-based BP to find the optimal weights of a 3-layer MLP for one hour ahead load forecasts using load time series and weather variables	3-layer MLP using BP and GA-BP	MAPE: GA-BP 1.6% (data calculated from results for day max load)
The comparison of mid term load forecasting between multi-regional and whole country area using Artificial Neural Network [56]	2010	To compare the forecasting results using MLP with data of Thailand as a whole or disaggregated in several regions.	MLP	MAPE monthly consumption multi-region: 1.45 peak: 2.48
Forecasting electricity demand in Thailand with an Artificial Neural Network approach [48]	2011	To compare MLP with ARIMA and Multi-Linear Regression for LTLF for Thailand using previous load time series and economical variables.	Different topologies of MLP and RBF.	MAPE MLP: 0.96%
A new neural network approach to short term load forecasting of electrical power systems [50]	2011	To compare performance of ANN using MHS (Modified Harmony Search) learning algorithm with other techniques STLF forecast using PJM ISO data	ARMA, RBF, MLP trained by BR (Bayesian Regularization), MLP trained by BFGS (Broyden, Fletcher, Goldfarb, Shanno) and MLP neural network trained by LM	MAPE: MLP MHS 1.39%
PREDICT – Decision support system for load forecasting and inference: A new undertaking for Brazilian power suppliers [41]	2011	To analyze the use of wavelets, time series analysis methods and artificial neural networks, for both mid and long term forecasts.	MLP with BP and LM	MAPE: 0.72%
Monthly electricity demand forecasting based on a weighted evolving fuzzy neural network approach [37].	2011	To compare WEFuNN (Weighted Evolving Fuzzy Neural Network) with ENN and BPN for 1-month ahead load forecast.	WEFuNN, Winter’s, MRA	MAPE WEFuNN: 6.43%
Short-term power load forecasting based on self-adapting PSO-BP neural network model [22]	2012	To show that PSO-BP algorithm can obtain optimal MLP parameters outperforming BP to forecast hourly 1-day ahead load demand for a city of China.	MLP getting the parameters with PSO-BP and BP	MAPE PSO-BP: 2.39%
A comparison of support vector machines and artificial neural networks for mid-term load forecasting [29]	2012	To compare the performance of SVM and ANN for MTLF with load and weather data.	MLP with several different numbers of neurons (2, 5, 8, 20/30). Usage of GA and PSO to obtain optimal SVMs models.	The authors conclude that both ANN and SVM are suitable, but SVM is more reliable and stable for load forecasting.
Load forecasting in a smart grid oriented building [15]	2013	To compare performance of ARIMA, MLP, SVM and STLF (next hour forecast) in university campus microgrid.	Seasonal ARIMA, MLP and SVM.	MAPE MLP: 5.3%
Short-term load forecasting for microgrids based on Artificial Neural Networks [46]	2013	To check ANN performance for load forecasting in a microgrid-sized Spanish region from previous load time series.	MLP (16 neurons in hidden layer)	MAPE: 2–5%
Multi-substation control central load area forecasting by using HP-filter and double neural networks (HP-DNNs) [42]	2013	To compare the use of HP (Hodris-Prescott) filter to decompose the previous load signals into trend and cyclical signals and DNN (Double Neural Network) for LTLF with other algorithms.	HP-DNN	MAPE HP-DNNS: 1.42–3.20%
Check the performance of MLP using SOM and k-means to find the right number of MLPs for STLF for a microgrid in Spain [35].	2014	To check the performance of MLP using SOM and k-means to find the right number of MLPs for STLF for a microgrid in Spain.	3-stage: SOM + k-means clustering and MLP. No other algorithms were tested.	MAPE: 2.73–3.22%
PI-controlled ANN-based energy consumption forecasting for smart grids [17].	2015	To compare ANN and PI-ANN (Proportional Integral ANN) to predict consumption of individual devices.	PI-ANN and MLP.	N/A
Short-term load cross-forecasting using pattern-based neural models [25]	2015	To check if a combination of daily and weekly patterns performs better than the models individually for SLTF from previous load.	Unspecified neural model	MAPE cross- forecasting: 0.85%
Input data analysis for optimized short term load forecasts [26]	2016	To compare the performance of MLP, SVR and clustering for 24-ahead forecast for Germany load demand.	MLP(1,1,1) with (LM) algorithm, SVR and k-means cluster.	MAPE SRV: 2.1%
Hourly load forecasting model based on real-time meteorological analysis [18]	2016	To check the influence of weather variables in load forecast using MLP.	3-layer MLP	MAPE (including weather variables) <2%
Neural network-based short-term electricity demand forecast for Australian states [27]	2016	To check the performance of FFNN (Feed Forward Neural Network) forecasting model for the different regions of Australia for STFL.	FFNN (using LM for training)	MAPE: 2.7233%
Building energy load forecasting using deep neural networks [19]	2016	To compare standard LSTM and LSTM-based Sequence to Sequence for STFL for 1-min resolution 1-h ahead predictions.	LSTM and LSTM-based S2S.	RMSE LSTM-S2S: 0.667
Deep neural network-based demand side short term load forecasting [21]	2016	To compare DNN forecasting results for individual industrial consumers from South Korea with typical 3-layered shallow neural network (SNN), ARIMA, and Double Seasonal Holt-Winters (DSHW) model.	DNN (4 hidden layers with 150 neurons per layer and using RBM and ReLU), ARIMA, DSHW, MLP	DNN RBM: MAPE 8.84% RRMSE 10.62%
Forecasting daily electricity load by wavelet neural networks optimized by Cuckoo search algorithm [71]	2017	To check performance of MLP using wavelet for data-preprocessing and Cuckoo algorithm to obtain parameters.	MLP (using Wavelet and Cuckoo algorithm), ARIMA, MLR	MAPE Wavelet ANN-CS: 0.058
Short-term forecasting of electricity demand for the residential sector using weather and social variables [58]	2017	To compare algorithms to forecast 1-month ahead demand in South Korea.	SVR, Fuzzy-rough feature selection with PSO, MLP, MLR and ARIMA.	MAPE SVR fuzzy-rough: 2.13%
A comparison of artificial neural networks and support vector machines for short-term load forecasting using various load types [28]	2017	To compare SVM and ANN to predict the load of Trinidad and Tobago for 3 industrial customers with different consumption patterns: continuous, batch, batch-continuous.	3-layer MLP and SVM.	MAPE ANN: 1.04%
Short-term load forecasting using EMD(Empirical Mode Decomposition)-LSTM neural networks with a Xgboost algorithm for feature importance evaluation [51]	2017	To compare SD (Similar Days)-EMD-LSTM algorithm with others used for STLF.	SD-EMD-LSTM, LSTM SD-LSTM EMD-LSTM, ARIMA, BPNN, SVR	MAPE SD- EMD- LSTM 24 h: 1.04% 168 h: 1.56%
Deep learning for household load forecasting—A novel pooling deep RNN [20]	2018	To compare the performance of PDRNN (Diagonal Recurrent Neural Networks) with other algorithms for STLF household forecast.	PDRNN with ARIMA, SVR, DRNN, SIMple RNN.	MAPE PDRNN: 0.2510%
Long short term memory networks for short-term electric load forecasting [30]	2017	To compare algorithms for STLF regional load forecasting.	LSTM, MLP, ARIMA.	MAPE LSTM: 3.8%
A State-of-the-Art Review of Artificial Intelligence Techniques for Short-Term Electric Load Forecasting [3]	2017	To compare performance of ANFIS, MLP and SVM for STLF in a large region.	MLP, SVM and ANFIS	MAPE SVM: 1.790%
Short term load forecasting using deep neural networks (DNN) [72]	2018	To compare different transfer functions using MLP for STFL in an Iberian region.	MLP using different transfer functions: sigmoid, ReLU and ELU.	MAPE MLP ELU-ELU: 2.03%
Residential load forecasting using deep neural networks (DNN) [52]	2018	To compare DNN algorithms for STFL day-ahead for residentials users.	LSTM, GRU, RNN, ARIMA, GLM, RF, SVM, FFNN.	MAPE LSTM: 29%
Optimal deep learning LSTM model for electric load forecasting using Feature Selection and Genetic Algorithm: Comparison with Machine Learning Approaches [47]	2018	To find optimal algorithm for STLF and MTLF for region load, using GA to find optimal parameters.	LSTM + GA, Ridge Regression, Random Forest, Gradient Boosting, Neural network, Extra Trees.	RMSE LSTM 0.61%
Predicting electricity consumption for commercial and residential buildings using deep recurrent neural networks [43]	2018	To evaluate an LSTM-based algorithm using MLP for encoding for MTLF of different residential building load profiles.	LSTM + MLP + SMBO	N/A
Predicting electricity consumption using deep recurrent neural networks [53]	2019	To compare RNN and LSTM to predict load in STLF MTLF and LTLF.	RNN, LSTM, ARIMA, MLP, DNN	ARIMA for STLF RNN and LSTM for MTLF and LTLF.
Short-term load forecasting in grid-connected microgrid [14]	2019	To compare the performance of algorithms for STLF in microgrid.	GMDH, MLP-LM	RMSE MLP: 0.062%
Short-term load forecasting at different aggregation levels with predictability analysis [54]	2019	To compare different algorithms for STLF at different aggregation levels.	MLP, LSTM, GBRT, Linear regression, SVR	N/A
Short-term residential load forecasting based on LSTM recurrent neural network [31]	2019	To compare the performance of forecast algorithms depending on the level of aggregation of AMI data.	LSTM+BPNN variants, KNN and mean.	MAPE LSTM: ind 44.39%, aggregated forecast: 8.18%, forecast aggregation: 9.14%
Day-ahead prediction of microgrid electricity demand using a hybrid Artificial Intelligence model [49]	2019	To compare different optimization algorithms before using FFANN for STLF using load and economic input variables.	SA-FFANN, WT-SA- FFANN, GA-FFANN, BP-FFANN, (PSO)-FFANN	MAPE WT-SA -FFANN: 2.95%
Electricity consumption probability density forecasting method based on LASSO-Quantile Regression Neural Network [45]	2019	To compare LASSO-QRNN for electricity consumption probability density LTLF	LASSO-QRNN	MAPE LSTM: 0.02%
Forecasting electric load by aggregating meteorological and history-based Deep Learning modules [32]	2020	To compare the combination of LSTM and DNN for STLF with LSTM alone.	LSTM+DNN, LSTM and DNN	MAPE LSTM+DNN: 4.28%
A Deep Learning approach to forecasting monthly demand for residential–sector electricity [44]	2020	To compare LSTM with other algorithms for MTLF.	SVR, MLP, ARIMA, MLR, LSTM	MAPE LSTM: 0.07%

## Data Availability

All the papers included in this systematic review have been downloaded from the original publisher sites and are available to any researcher. Some of the publisher platforms may require a subscription to be able to read and download the papers.

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
