# Peer review of "Systematic Review of Electricity Demand Forecast Using ANN-Based Machine Learning Algorithms"

_sensors, 2021, doi:10.3390/s21134544_

Round 1

Reviewer 1 Report

In this paper, a systematic review of electricity demand forecast using ANN-based machine Learning algorithms is given. Extensive literature summaries are provided. However, the paper does not summarize the specific methods and strategies.The advantages and disadvantages of a certain class of methods, as well as the applicable scenarios, need to be explained. In addition, schematic block diagrams or flow charts of the widely used methods need to be provided.

Author Response

Thanks for your valuable feedback.

We have added more content to section 4, including a comprehensive flow chart that shows a generalization of the process followed by the authors, from the raw data to the presentation of results. This diagram is explained in detail in a new subsection of section 4.

The idea of including diagrams of the best performing models was discarded, because there are many different ones that attain similar performance. Many authors use algorithm variants and combinations of different algorithms so we found it hard to include all this information in an article. We preferred to focus on providing an extensive bibliography and precise references so that the reader can access the information easily. 

We had the paper proofread by a native English speaker, who pointed out a number of typos and grammar errors; he also suggested rewriting some sentences to convey the same meaning in a lighter and more straightforward manner. 

Reviewer 2 Report

This paper reviews the various methods of using artificial neural networks (ANN) to predict power demand. The most relevant papers are selected, then the prediction range, input variables, output variables, measuring and comparing performance, origin of the training data and other categories are discussed in detail. The results obtained by these algorithms, as well as the verification and comparison of the results, highlight some important and effective features and algorithm configurations. At the same time, the number of papers of different categories is also listed for intuitive display. Therefore, to some extent, this paper provides help and inspiration for new scholars and experienced researchers.

Below I enclose a more detailed review of the manuscript.

(1) “4.2. Non-linearity with respect to input variables”, “4.4. Input variables”, and “4.5. Pre-processing of input variables” can be combined because they are all related to input variables. In addition, the pre-processing of input variables is not comprehensive enough and needs to be supplemented.

(2) The structure of the conclusion is unclear and confusing. It would be better if a corresponding summary could be made based on the analysis in Section 4. In addition, the paper is not professional enough in some language expressions, there is still room for improvement.

(3) The paper lists the relevant papers from ScienceDirect, Scopus and IEEE Xplore. But if possible, please add more latest paper about ANN and power load into references. For example, Yaoyao He; Rui Liu; Haiyan Li; Shuo Wang; Xiaofen Lu; Short-term power load probability density forecasting method using kernel-based support vector quantile regression and Copula theory, Applied Energy, 2017,185:254-266 could be used in Section 4.1, and Yaoyao He; Yang Qin; Shuo Wang; Xu Wang; Chao Wang; Electricity consumption probability density forecasting method based on LASSO-Quantile Regression Neural Network, Applied Energy, 2019, 233-234: 565-575 can be added to Section 4.2. Besides, references Yaoyao He; Yun Wang; Short-term wind power prediction based on EEMD-LASSO-QRNN model, Applied Soft Computing, 2021 ,105:107288 can be added to Section 4.7.

Reviewer 3 Report

The subject, the literature search and results of the specific paper are quite interesting.

Specific comments:

  • The term “power central” in page 7 line 256 does not provide a clear meaning. The authors are advice to change accordingly to provide a clear meaning.
  • The categorization in “Origin of the training data” does not seem quite meaningful. Probably “Aggregated” and “Not Aggregated” would fit better. Smart meters can be installed inside microgrids for acquiring data from the central point or individual loads so this depicts the weakness of the categorization chosen.
  • It will be nice to include some graphical results of the methods found to be performing well along with all the details of the algorithms, data analysis etc.

Round 2

Reviewer 1 Report

The author has responded well to most of the comments. Good work.

Reviewer 2 Report

The authors have adequately addressed all the comments that raised in a previous round of review.